# *Colletotrichum echinochloae*: A Potential Bioherbicide Agent for Control of Barnyardgrass (*Echinochloa crus-galli* (L.) Beauv.)

**DOI:** 10.3390/plants12030421

**Published:** 2023-01-17

**Authors:** Qiongnan Gu, Shihai Chu, Qichao Huang, Anan Chen, Lin Li, Ruhai Li

**Affiliations:** Key Laboratory of Integrated Pest Management on Crops in Central China, Ministry of Agriculture and Rural Affairs/Hubei Key Laboratory of Crop Disease, Insect Pests and Weeds Control/Institute of Plant Protection and Soil Fertilizer, Hubei Academy of Agricultural Sciences, Wuhan 430064, China; qiongnan.gu@foxmail.com (Q.G.); chushihai1@163.com (S.C.); qichao.huang@foxmail.com (Q.H.); hmt202301@163.com (A.C.); sbss_01@163.com (L.L.)

**Keywords:** biocontrol, mycoherbicide, *Colletotrichum*, barnyardgrass

## Abstract

Barnyardgrass (*Echinochloa crus-galli* (L.) Beauv.) is one of the most troublesome weeds in transplanted and direct-seeded rice worldwide. To develop a strategy for the biocontrol of barnyardgrass, fungal isolates were recovered from barnyardgrass plants that exhibited signs of necrosis and wilt. An isolate B-48 with a high level of pathogenicity to barnyardgrass was identified after pathogenicity tests. From cultural and DNA sequence data, this strain was identified as *Colletotrichum echinochloae*. The inoculation of the barnyardgrass plant with *C. echinochloae* caused a significant reduction in fresh weight. The isolate B-48 was highly pathogenic to barnyardgrass at the three- to four-leaf stages. When inoculated at a concentration of 1 × 10^7^ spores/mL, barnyardgrass could achieve a reduction in fresh weight of more than 50%. This strain was safe for rice and most plant species. The results of this study indicated that this strain could be a potential mycoherbicide for barnyardgrass control in paddy fields in the future.

## 1. Introduction

*Echinochloa crus-galli* (L.) P. Beauv., commonly known as barnyardgrass, is an annual summer plant found in a variety of different soils and climates [1,2,3]. It is a widely distributed weed species and is recognized as the world’s most serious weed of rice, affecting up to 36 types of crops in 61 countries [1,4]. It can proliferate in temperate, tropical, and subtropical regions, and a single barnyardgrass plant can produce up to 39,000 seeds [5,6]. Due to its wide ecological tolerance, high adaptability, rapid germination, abundant seed production, and strong competitive ability, economic losses from its proliferation continue to mount [7,8,9,10]. This noxious weed can cause yield losses between 21 and 79% in rice, depending on the cropping system and management [11].

In agriculture, weed control is achieved mainly through herbicides because of their effectiveness against most weeds. Unfortunately, synthetic chemicals may have adverse effects on humans as well as on soil, aquatic, and terrestrial fauna and flora; they pollute the environment and contribute to the emergence of new problematic weeds [12,13,14,15]. Furthermore, the repeated use of herbicides has led to the evolution of herbicide-resistant weed populations [16,17]. *Echinochloa* spp. are weedy plants that have evolved resistance to several herbicide classes [18,19,20,21]. To minimize the use and risk of herbicides, reducing herbicide doses in chemical weed control has been extensively investigated in recent years and has provided sufficient weed control and desirable yield [22].

In parallel, biological control has become a promising alternative for application in agriculture in recent years [23]. Due to its specificity, longevity and ability to overcome weed resistance to chemical herbicides, this control method has the benefit of being completely compatible with the environment [24]. Among these, one of the most promising branches of biological control is mycoherbicides, and at least 15 products are currently commercialized worldwide [25]. Examples include BioMal, a formulation of *Colletotrichum gloeosporioides* f.sp. *malvae*, introduced for the control of round leaf mallow (*Malva pusilla*) [26,27], and *C. gloeosporioides* f.sp. *aeschynomene*, which was released for the control of northern jointvetch (*Aeschynomene virginica*) in the United States in 1982 as Collego and again in 2006 as LockDown (EPA Registration Number 82681-1) [28].

Several other species within the genus *Colletotrichum* have been investigated as potential mycoherbicide agents [29,30]. Additional examples include Coltru, as the principal biocontrol agent in *C. truncatum*, which was investigated to control hemp sesbania (*Sesbania exaltata*) [31]. *C. orbiculare* was investigated for its potential to control spiny cocklebur (*Xanthium spinosum*) [32,33]. *C. gloeosporioides* f. sp. *cuscutae* was investigated as the commercial product Lubao, which was investigated to control *Cuscuta* spp. [34]. Hatatak is a formulation of *C. gloeosporioides* which was developed to control *Hakea sericea* [35]. Genome mining of two *Colletotrichum* species of *C. gloeosporioides* and *C. orbiculare* found that both species contained several pathogenesis genes which were important for pathogen infection [36]. There is also evidence that both *Colletotrichum* species have the ability to produce the plant hormone indole acetic acid, which is a well-established herbicide template [37]. Evidence suggests that this genus has a high potential for developing new mycoherbicides.

Wetness duration, dew period temperature, plant growth stage, pathogen concentration, and adjuvants are the major factors associated with bioherbicide efficacy [38,39,40,41]. A sufficient source of infection is the first step to causing an infection of the target host, so the appropriate concentration of the pathogen is essential for the pathogen to colonize the host. An appropriate duration of wetness (humidity) and temperature of the dew period are the prerequisites for the germination and establishment of the bioherbicide agent in weeds [42], while a long duration of wetness is essential for infection on the aerial surfaces of the target host [43]. Often, inoculation at the early stage of weed growth will result in the control of weed infestations; usually, the old leaves of the plant have more lignin, which makes the pathogen more difficult to colonize on the host. Liquid formulations of mycoherbicides are the best for weed control and are used primarily to incite leaf and stem diseases [44]. A study conducted by Shabana showed that an *Alternaria eichhorniae* spore formulation based on vegetable oil (cotton seed oil and sunflower oil) to control water hyacinth significantly increased the disease severity compared with the untreated control [45]. Similarly, Boyette et al. proposed that the use of the inverted oil emulsions could be used to check moisture loss, and this tactic improved *C. truncatum*’s 100% efficacy for the control of *S. exaltata* [46].

Here, we demonstrated the potential mycoherbicide agent derived from an isolate of *Colletotrichum echinochloae*, identified from barnyardgrass weeds in Hubei Province, China and evaluated the biocontrol efficacy of the fungus for barnyardgrass in controlled conditions.

## 2. Results

### 2.1. Isolation and Identification of Fungal Pathogens for the Control of Barnyardgrass

The isolate of *C. echinochloae* B-48 was obtained from diseased barnyardgrass plants in a paddy field in Hubei Province, China (32°7′49″ N, 112°20′35″ E, 91 m above sea level) in 2018. This strain was isolated through single-spore isolation, and its pathogenicity on barnyardgrass was confirmed by the Koch’s postulates.

Unlike most *C. echinochloae* strains, isolate B-48 produced pink mycelium in PSA medium (Figure 1b). PDA colonies of *C. echinochloae* are hyaline and usually form black acervuli (Figure 1a). After inoculation of strain B-48 for 14 days, black irregular-shaped sclerotia formed and were visible on the surface of the PSA medium. The sclerotia’s range was 0.3–8.5 × 0.5–9.5 μm (avg. 3.5 × 5.7 μm). The conidia of strain B-48 harvested from PSA were oval, often guttulate, with a size of 10.7–14.3 × 3.8–5.5 μm (avg. 12.3 × 4.7 μm, 300 conidia) (Figure 1c). The conidia harvested from the host plant were falcate, their apices acute or sharply acute, usually guttulate, with a size of 14.7–17.2 × 3.6–5.2 μm (avg. 15.8 × 4.8 μm, 300 conidia). The hyphae were colorless, septate, usually guttulate, with a size of 1.0–5.8 μm. The appressoria were globose to perprolate, ovoid, obovoid, or clavate, smooth, lobate, or multilobate, their apices cylindrical or obtuse, their edges entire, with a size of 8.2–14.8 × 7.5–12.3 μm, (avg. 12.2 × 11.7 μm, 300 appressoria). The setae were dark brown, septate, rounded at the base, their apices acute, with a size of 6.2–8.7 × 63.7–114.0μm (Figure 1d). These morphological features were similar to the reference *C. echinochloae* [47,48].

For the molecular identification of strain B-48, genomic DNA was isolated with the fungal genomic DNA extraction kit (Solarbio, Beijing, China). The internal transcribed spacer region (ITS), superoxide dismutase 2 (Sod2), DNA lyase gene (Apn2), and mating type protein (Mat1) were amplified, respectively. Based on the combined ITS, Sod2, Apn2, and Mat1 sequences, B-48 belonged to the same clade as the reference isolates of *C. echinochloae* (Figure 2).

### 2.2. Host-Range Studies

Both plant phylogeny and the economic importance of nontarget crops in Hubei Province were considered when determining the host range. Important field crops such as rice, corn, and wheat, economic crops such as cotton and tobacco, and vegetables such as chili and leek were examined for the host range (Table 1). In our experiment, the strain B-48 of *C. echinochloae* was safe for large-acreage field crops, economic crops, and vegetables grown in Hubei Province, China. However, strain B-48 caused a hypersensitive reaction on two rice cultivars, which showed necrotic specks (1 mm in diameter) 4 to 5 days after inoculation. However, the expansion was restricted to a slight enlargement or coalescence even after three weeks postinoculation. The reisolation of the strain could not be pursued.

Aside from barnyardgrass, which has a higher virulence, 25 other weedy plants were tested for the host range. All the plant species tested, except *Cyperus rotundus* L., were immune to this fungus, with no symptoms observed three weeks after inoculation (Table 2). Plant vigor and fresh weight were similar to the corresponding controls.

Based on the result of this experiment, *C. echinochloae* B-48 has a very limited range of hosts and only causes severe infections on barnyardgrass. However, it should be applied considerably in rice fields.

### 2.3. Effect of Different Temperatures on C. echinochloae Conidial Germination, Appressorial Formation, and Mycelial Growth

In the present study, the effects of temperature on conidial germination, appressorial formation, and mycelial growth were examined. The optimum temperature for conidial germination was between 25 and 30 °C (Figure 3a). The highest germination rate was observed at 30 °C. The conidia did not germinate below 5 °C. The appressorial formation showed a similar trend to the conidial germination (Figure 3a). The appressorial formation was consistently lower below 15 °C, but the highest appressorial formation rate was observed at 30 °C. In general, 30 °C may be the best temperature for conidial germination and appressorial formation.

Mycelial growth for strain B-48 of *C. echinochloae* was also examined at different temperatures (Figure 3b). The optimal temperature range for growth was 15–30 °C. Mycelial growth was the fastest at 30 °C, relatively fast at 15–25 °C, but slower at 10 °C or less and at 35 °C or more.

These results suggest that the best temperature for conidial germination, appressorial formation, and mycelial growth was between 25 and 30 °C, which suggests that infection may occur throughout the barnyardgrass growth stage.

### 2.4. Effect of C. echinochloae Conidia Concentration on Weed Control

To induce maximum disease destruction on the weed, we applied varying inoculum concentrations of the *C. echinochloae* isolate B-48 on barnyardgrass. The severity of the disease in barnyardgrass resulted in significant differences. At a concentration of 1 × 10^4^ and 10^5^ spores/mL, the fungus caused few sporadic lesions; the severity of the disease was only 8% and 27%, and the fresh weight of the plant was not affected. When plants were inoculated with spore concentrations greater than 10^6^ spores/mL, plants achieved a reduction in fresh weight of more than 50%. The LD50 value was also standardized and found to be 10^6^ spores/mL. In comparison, efficacies increased most dramatically when inoculated with spore concentrations of 10^7^–10^8^ spores/mL. At 10^7^ and 10^8^ spores/mL, the plant fresh weight inhibition rate could reach 78% and 84% (Table 3).

### 2.5. Effect of Weed Growth Stage on Biocontrol Efficacy

Different stages of barnyardgrass growth were tested for biocontrol efficacy. The result showed that the seedlings were more sensitive to *C. echinochloae* (Figure 4). Disease symptoms appeared 72 h after inoculation in plants in the 3–4-leaf stage. After 7 days postinoculation, symptoms began as light-dark to dark brown lesions and then progressed to water-soaked areas on stems and foliage, and finally to necrosis of all tissues. The plants collapsed with few of the new leaves’ recovery. The inhibition rate of three- to four-leaf stage plant was 79% at 3 weeks after inoculation. The efficacy was comparable to the five- and six-leaf stages. The plants showed disease symptoms 72 h after inoculation, and the inhibition rates were 57% and 56% at 3 weeks after inoculation. The effectiveness was noticeably lower in plants at the seven-, eight-, and nine-leaf stages, especially with fresh weight reduction. The plants at the seven-, eight-, and nine-leaf stages treated with the strain B-48 all had remaining live leaves, and all the plants recovered. The inhibition rates for the seven-, eight-, and nine-leaf stage plants were less than 30%, which indicates that *C. echinochloae* B-48 is not a supervirulent strain in barnyardgrass (Table 4).

### 2.6. Effect of Dew Period on Biocontrol Efficacy

The infection of barnyardgrass by *C. echinochloae* B-48 was significantly influenced by the dew period. In general, disease severity increased dramatically in response to the duration of leaf wetness, especially during the first 10 h after inoculation. The pathogen established only a light infection with a 6 to 8 h dew period. When the dew period was less than 6 h, no symptoms were observed. The infection increased most rapidly with leaf wetness from 12 h, resulting in an increase in the severity of the disease to 82% and a reduction in the fresh weight of the plant to approximately 68% (Table 5).

## 3. Discussion

Many species in the genus *Colletotrichum* have been reported to be effective biocontrol agents against weeds [49]. These included *C. gloeosporioides* f. sp. *Malvae* (effective against *Malva pusilla* and *Malva parviflora*) [26,27], *C. gloeosporioides* f. sp. *aeschynomene* (effective against *Aeschynomene virginica*, *A. indica*, and *Sesbania exaltata*) [31], *C. coccodes* (effective against *Abutilon theophrasti*) [50,51], *C. dematium* (effective against *Parthenium hysterophorus*) [52], *C. orbiculare* (effective against *Xanthium spinosum*) [32,33], *C. truncatum* (effective against *Sesbania exaltata*) [53], etc. Among these species, only Collego (LockDown), BioMal, Velgo, and Lubao1 have been successfully commercialized. This genus of fungi has been reported to control various weeds around the world and has excellent potential to develop into a commercial mycoherbicide [54]. Here, in this study, we demonstrated that *C. echinochloae* had the potential to be developed as a mycoherbicide for the management of barnyardgrass (*Echinochloa crus-galli* (L.) Beauv.).

Our studies showed that the inoculation of the barnyardgrass plant with *C. echinochloae* strain B-48 caused a significant reduction in the fresh weight of barnyardgrass. After 1–3 weeks postinoculation, *C. echinochloae* isolate B-48 effectively controlled barnyardgrass with 60–80% disease incidence, and the fresh weight inhibition rate could reach 80%. The laboratory experiment revealed that *C. echinochloae* had a narrow host range. It showed no pathogenicity against other economically important crops in Hubei, including field crops, economic crops, and vegetables. The fungus only caused an HR response in two rice cultivars. To provide a more accurate assessment of pathogenicity, additional research on this fungus in a wider variety of rice cultivars may be required.

In previous study, fungal taxa recorded solely on one host does not imply that they are host-specific. Phylogenies for the falcate-spored graminicolous *Colletotrichum* revealed a close link between morphological groupings. *C. echinochloae* is originated from the same host plant as *C. jacksonii*, and while the two species are clearly sister taxa. However, the type strains of *C. echinochloae* and *C. jacksonii* fall into two distinct phylogenetic lineages [55]. In our experiment, *C. echinochloae* was pathogenic to *E. crus-galli*, producing leaf blight and grayish-white lesions with brown surroundings in the leaves. It did not cause necrotic lesions in *Z. mays*. *C. graminicola* infects only *Z. mays* and never infects *E. utilis* [56]. These comparable results were reported previously [56]. The morphological characteristics were not enough for differentiating B-48 from *C. echinochloae* and *C. jacksonii.* According to other studies, using phylogenetic analysis is better to understand species delimitation. Crouch et al. [47] stated that phylogenetic species delimitation in of the falcate-spored graminicolous *Colletotrichum* should be based on the combined ITS, Sod2, Apn2, and Mat1 gene regions. So, using both morphological and combined phylogenetic analysis, B-48 was finally identified as *C. echinochloae.*

Bioherbicidal activity depends on fungal attachment, germination, and penetration of the host to initiate symptoms. *C. echinochloae* infection begins with conidia attachment to the leaf surface, followed by germ tubes that germinate and quickly differentiate into melanized appressoria as with many other *Colletotrichum*. The melanized appressoria accumulate significant turgor pressure and help the pathogen penetrate the host plant cell wall. Following penetration, biotrophic primary hyphae and necrotrophic secondary hyphae colonize the host cells [57]. Temperature and relative humidity are the key factors for conidial germination, appressorial formation, and infection. In our study, the temperature range of 5 to 30 °C observed for conidial germination and appressorial formation was similar. This was similar to other pathogens in which the optimum temperature for appressorial formation coincides with conidial germination [58]. For example, the optimum temperature for *C. gloeosporioides* conidial germination and appressorial formation was similar after 24 h of incubation [59]. Furthermore, the optimal temperature range (15–30 °C) for conidial germination and appressorial formation of *C. echinochloae* suggests that infection may occur throughout the growth stage of the barnyardgrass. And what’s more, our experiment showed that increased duration of dew period resulted in increased incidence of infection. This interaction of temperature and wetness duration is common among foliar-applied mycoherbicide agents [60]. In summary, the best time for the application of *C. echinochloae* for the biocontrol of barnyardgrass would be in the summer and with high humidity conditions.

The requirement for a relatively high inoculum concentration is a key factor in infection [61]. Conidia are considered the most infectious of fungal propagules [62] and, therefore, the most suitable form for bioherbicide formulations. In this study, we applied *C. echinochloae* with a conidial suspension in Tween-80 and corn oil. Tween and oil could help increase the fungal attachment and relative humidity. When applied with concentrations close to 10^7^ spores/mL, the plants reached an 80% fresh weight inhibition. However, the disease incidence and fresh weight inhibition rate of inoculation with 10^7^ and 10^8^ spores/mL were similar. This may be for two reasons: one is that *C. echinochloae* is not a supervirulent strain on barnyardgrass. The other is that *C. echinochloae* has two forms of conidia; falcate conidia have been reported to be more efficient on host leaves than oval conidia [63]. For this reason, it may be prudent to continue searching for *C. echinochloae* variants with a higher virulence or screen the best medium for harvesting higher-virulence falcate-shaped conidia. Our findings suggest that the fungal dose is critical to the effectiveness of controlling barnyardgrass, and this notion fits the general pattern of infection by plant pathogens [64]. The effective dose range identified is similar to those reported with other foliar bioherbicide agents [65,66,67], and the requirement for high inoculum doses may represent a potential economic limitation for weed control in the field.

The *C. echinochloae* strain B-48 succeeded in killing the three to four true leaf seedlings, and very few of them had a remaining live leaf. On plants at the four-leaf stage or younger, the pathogen caused a disease severity greater than 80% and more than 79% fresh weight reduction compared to the respective controls in each growth stage. For barnyardgrass at the five-leaf stage and six-leaf stage, plants were also severely damaged after inoculation, but few of the plants recovered. For plants over the seven-leaf stage, especially large plants, lesions on emerging young leaves expanded slowly, rarely coalesced, and rarely caused complete plant mortality. The lower efficacy on older plants was likely due to significantly less disease on the youngest leaf that continued to support weed regrowth, which diminished the fresh weight reduction originating from the damage on the lower leaves. The low disease severity of pathogen to old plants is common among foliar-applied mycoherbicide agents [60].

In conclusion, our findings suggest that *C. echinochloae* isolate B-48 has mycoherbicidal potential for barnyardgrass. Further research is therefore needed and should focus on inoculum and formulation performance for practical use and enhancing weed suppressive activity through integration with feasible and cost-effective weed control practices.

## 4. Materials and Methods

### 4.1. Isolation, Identification, and Characterization of C. echinochloae B-48

*C. echinochloae* isolate B-48 was obtained from diseased barnyardgrass plants in a paddy field in Hubei Province, China (32° 7′ 49″ N, 112° 20′ 35″ E, 91 m above sea level) in 2018. Its pathogenicity on barnyardgrass was confirmed by the Koch’s postulates. The isolate was reinoculated on barnyardgrass and reisolated from leaf lesions on the host. The isolate was stored on potato dextrose agar (PDA) at 4 °C at the China Center for Type Culture Collection (CCTCC) with M2020689. The original identification of the fungus was based on the disease symptoms and fungal morphology.

For molecular identification, total genomic DNA was isolated from mycelia collected from 7-day-old colonies of B-48 using the fungal genomic DNA extraction kit (Solarbio, Beijing, China). As previously described [47,48,68,69], the internal transcribed spacer region (ITS), superoxide dismutase 2 (Sod2), DNA lyase gene (Apn2), and mating type protein (Mat1) were amplified by pairs ITS1/4, SOD625F/R, Apn1W1F/Apn1W1R, Mat1M72F/Mat1M72R, respectively. The sequence results were analyzed using the web-based blasting program, the basic local alignment search tool (BLAST), and the data were compared with the GenBank database at the National Center for Biotechnology Information (NCBI) nucleotide sequence database. A maximum parsimony analysis was performed on the multilocus alignment (ITS, Sod2, Apn2, Mat1-2) with PAUP (Phylogenetic Analysis Using Parsimony) v. 4.0 b10 [70] using the heuristic search option with 100 random sequence additions. The evolutionary history was deduced using the neighbor-joining method with a bootstrap test of 1000 replicates to determine the percentage of related taxa clustered together. The maximum composite likelihood method was used to calculate the evolutionary distances, which were used to infer the phylogenetic tree.

### 4.2. Plant Preparation, Pathogen Inoculum Preparation, and Plant Inoculation

The mature barnyardgrass seeds were harvested from our lab. In a 10 cm plastic container, about 15 seeds were placed in the growing medium. After being hydrated, seeded pots were placed in the greenhouse at 28 °C with a 16 h photoperiod. The barnyardgrass plants were grown within about three weeks until they had three to four leaves. For host-range studies, 5 to 15 seedlings were seeded in the growing medium and grown under identical experimental conditions for 3 to 5 weeks until the plants passed the three-leaf growth stage.

For the inoculum preparation, fungal cultures were transferred to a potato sucrose broth (PSB) medium and shaken at 220 rpm at 28 °C in dark conditions for approximately 7 days to achieve sporulation. The conidia suspension was filtered through four layers of cheesecloth to remove large fragments of mycelium. Concentrations of spore suspensions were estimated with a hemacytometer and adjusted for relative concentration.

The spore suspension was evenly sprayed onto barnyardgrass plants, crop plants, and weeds for plant inoculation. The inoculated plants were placed in a dark dew chamber at 28 °C for 24 h with almost 100% relative humidity and then transferred to a growth chamber at 25 °C with a photoperiod of 16 h. Lesion formation was examined after 7 days of incubation. All of these experiments were replicated three times.

### 4.3. Disease Assessment

The disease severity on all leaves was used primarily to evaluate the treatment’s effectiveness. The severity of disease symptoms was recorded using an index ranging from 1 (healthy leaf), 2 (chlorosis), 3 (small lesions < 2 cm), 4 (several lesions per leaf), 5 (big lesions > 2 cm) to 6 (dead leaf). In host-range studies and the biological assessment of *C. echinochloae*, the plant’s fresh weight was also used to determine the treatment effect. All plants in a pot were cut and weighed. The fresh weight inhibition rate (FWIR) was calculated using the following formula: FWIR (%) = (average FW of controls—average FW of pathogen-inoculated replicate)/average FW of controls ×100. Disease severity was assessed 7 days after inoculation, and fresh weight ratings were determined 21 days (3 weeks) after inoculation unless otherwise specified.

### 4.4. Host-Range Studies

The conidial suspensions of *C. echinochloae* B-48 at a concentration of 1 × 10^6^ spores/mL were prepared using the same application method as in the previous experiment. Under laboratory conditions, the pathogenicity assay was carried out on common cultivars of three cereal crops (wheat, rice, and corn), other vegetables, and crops grown widely in Hubei Province, China. Conidial suspensions of 1 × 10^6^ spores/mL in sterile distilled water were placed on test leaves, then incubated at 28 °C for 7 days under a high-humidity condition. Disease severity was measured 1 week after inoculation to assess the treatment effect.

To verify the host range of the isolate on other weeds, the inoculation of *C. echinochloae* was carried out on weedy plants of common summer species found in Hubei Province, China, including *Amaranthaceae*, *Chenopod*, *Convolvulaceae*, *Malvaceae*, *Polygonaceae*, *Portulacaceae*, *Rubiaceae*, *Scrophulariaceae*, *Solanaceae*, *Violaceae*, *Asteraceae*, *Commelinaceae*, *Cyperaceae*, and *Gramineae*. Ten leaves of each plant species were inoculated by the 1 × 10^6^ spores/mL suspension and were kept in a humid chamber at 28 °C with 100% relative humidity. Disease severity was measured 1 week after inoculation to assess the treatment effect.

### 4.5. Effect of Temperature on C. echinochloae Conidial Germination, Appressorial Formation, and Mycelial Growth

For conidia collection, strains were grown on PSB medium at 25 °C in dark conditions for 7 days. The conidia were harvested and suspended in sterile distilled water by centrifuging at 5000 rpm for 5 min and counted with a hemocytometer under a light microscope. For conidial germination and appressorial formation, 10 μL conidial suspensions at a concentration of 1 × 10^5^ spores/mL in sterile distilled water were placed on plastic coverslips and incubated at 5, 10, 15, 20, 25, and 30 °C for 48 h. Conidial germination and appressorial formation were observed and counted under the microscope. The percentages of conidial germination and appressorial formation were determined by microscopic examination of at least 300 conidia or appressoria. Mycelial growth was assessed by measuring colony diameter in plate cultures of *C. echinochloae* grown on PSA at 5, 10, 15, 20, 25, 30, 35, 40, and 45 °C for 7 days.

### 4.6. Effect of C. echinochloae Conidial Concentration on Weed Control

The liquid formulation was prepared by mixing corn oil, the fungal spore suspension (1:9, *v*/*v*), and 0.1% of Tween 80 [71]. The mixture was then adjusted to relative spore concentration and emulsified using a magnetic agitator for 3 min at low speed. Barnyardgrass plants in the three-leaf stage were inoculated with *C. echinochloae* B-48 formulation at spore concentrations of 10^4^, 10^5^, 10^6^, 10^7^, and 10^8^ spores/mL [65]. Four replicated pots were sprayed until runoff with suspensions of a relative concentration. Control plants were sprayed with water plus surfactant and oil at the same concentration. Disease severity and plant fresh weight were measured for the assessment of treatment effect.

### 4.7. Effect of Weed Growth Stage on Weed Biocontrol Efficacy

Barnyardgrass plants ranging from 3-to-9-leaf growth stages were tested for biocontrol efficacy. All the plants were inoculated with conidial suspensions of *C. echinochloae* B-48 at a concentration of 1 × 10^7^ spores/mL. The same application method was used as in the previous experiment. The control plants of each growth stage were sprayed with water plus surfactant and oil at the same concentration. The efficacy of weed control on plants of different growth stages was determined by comparing the disease severity and fresh weight of inoculated plants with their respective controls.

### 4.8. Effect of the Dew Period on Weed Biocontrol Efficacy

Conidial suspensions of *C. echinochloae* B-48 were prepared as previously described. The mixture was then adjusted to a concentration of 1 × 10^7^ spores/mL. Barnyardgrass plants at the three-leaf stage were inoculated and incubated in a humid chamber at 28 °C with 100% relative humidity for 0, 6, 8, 10, 12, and 24 h. Four replicated pots were sprayed until runoff with spore suspensions. Control plants were sprayed with water plus surfactant and oil at the same concentration. Disease severity and plant fresh weight were measured for the assessment of treatment effect.

### 4.9. Data Analysis

All experiments were carried out with three replicates per treatment and repeated twice in a completely randomized design. In inhibition rate assays, a log transformation of the inhibition rate data was used to normalize the distribution for further statistical analysis. Data were subjected to an analysis of variance (ANOVA), and when significant treatment differences were found (*p* < 0.05, or *p* < 0.01), means were compared using the least significant difference test (LSD).

## 5. Patents

This strain was submitted for a China invention patent and the application number is 202210802861.6.

## Figures and Tables

**Figure 1 plants-12-00421-f001:**
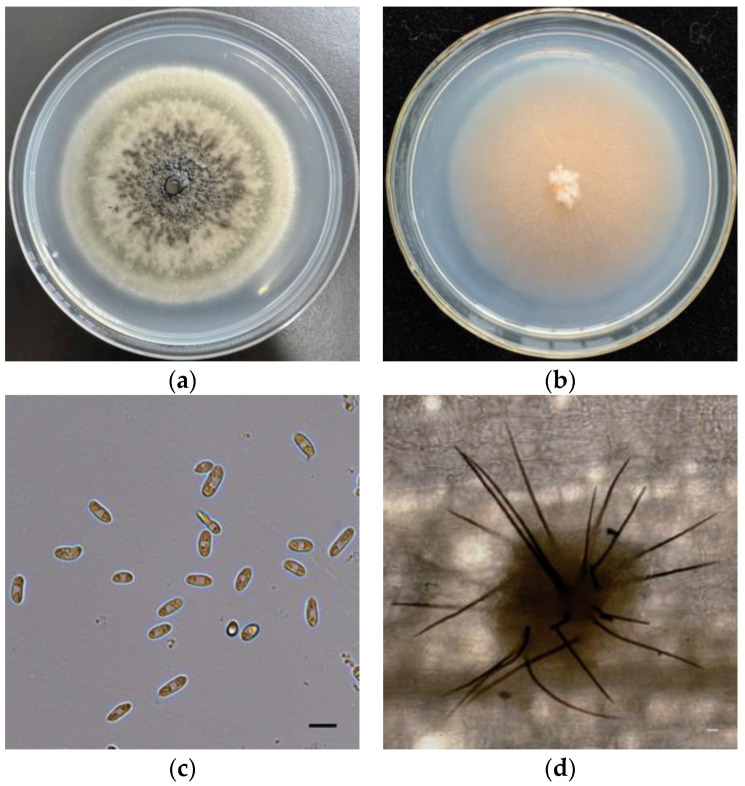
Morphology of *C. echinochloae* strain B-48. (**a**,**b**) Colony morphology cultured on potato dextrose agar (PDA) medium, potato sucrose agar (PSA) medium. (**c**) Morphology of conidia produced by PSA; bar scale: 10 µm. (**d**) Morphology of setae on barnyardgrass leaves; bar scale: 10 µm.

**Figure 2 plants-12-00421-f002:**
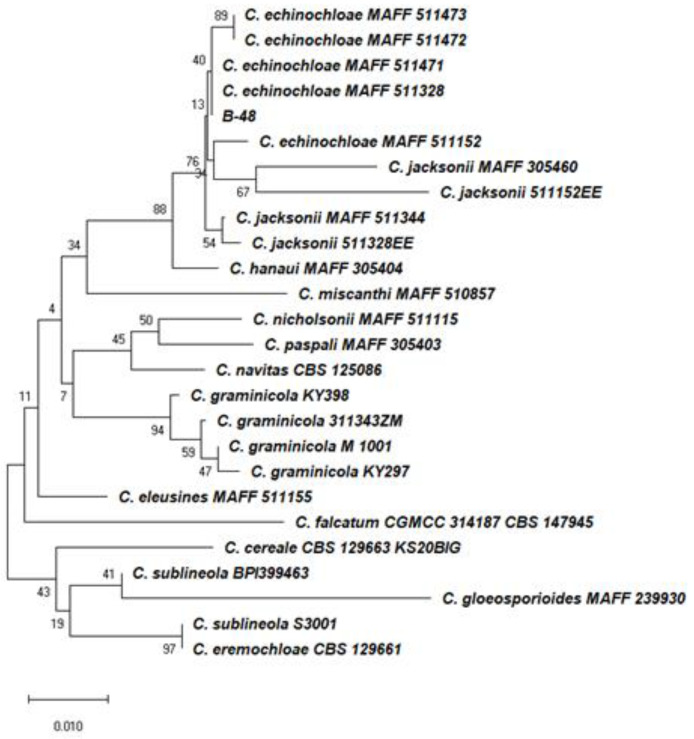
Phylogenetic tree showing the phylogenetic position of strain B-48 using the neighbor-joining (NJ) method based on concatenated ITS, Sod2, Mat1, and Apn2 gene sequence. The numbers at the nodes indicate the levels of the bootstrap support percentages based on the neighbor-joining of 1000 replicates. The scale bar represents a 0.01 sequence difference. The GenBank accession numbers for the fungal isolates used in the analysis are provided (Appendix A).

**Figure 3 plants-12-00421-f003:**
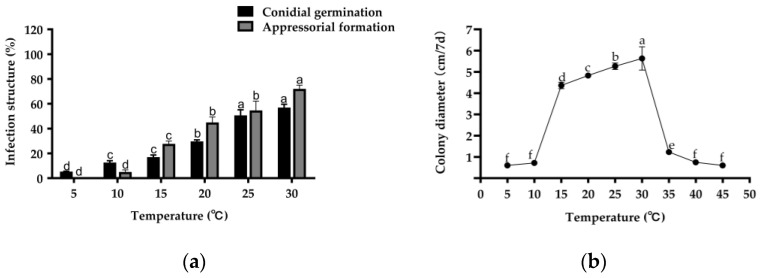
Effect of different temperatures on conidial germination of *C. echinochloae,* appressorial formation, and mycelial growth. (**a**) Effect of temperature on *C. echinochloae* B-48 conidial germination and appressorial formation. (**b**) Effect of temperature on *C. echinochloae* B-48 mycelial growth. The lower-case letter represents significant difference at *p* < 0.05.

**Figure 4 plants-12-00421-f004:**
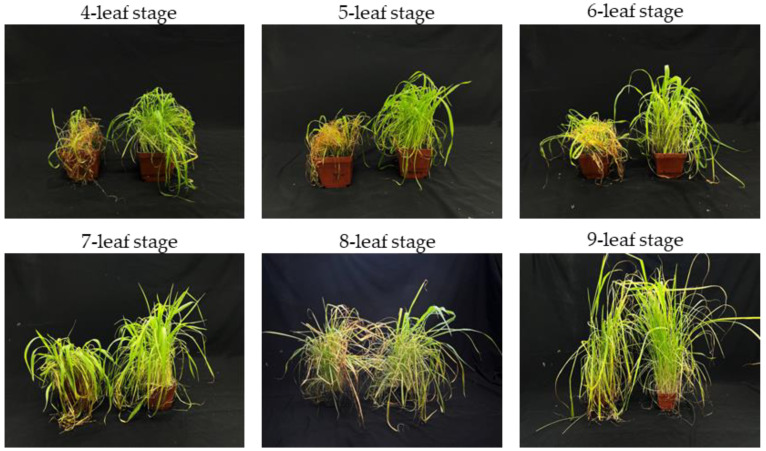
Effect of *C. echinochloae* B-48 on different weed growth stages.

**Table 1 plants-12-00421-t001:** Host susceptibility of *Colletotrichum echinochloae* in major summer crops and plants in Hubei Province.

Family	Species Treated	Plant Reaction
*Gramineae*	Maize	I
	Rice	HR
	Wheat	I
*Malvaceae*	Cotton	I
*Solanaceae*	Tobacco	I
	Chili	I
*Liliaceae*	Leek	I
*Dioscoreaceae*	Chinese yam	I
*Fabaceae*	Cowpea	I

I, plants are immune and showed no visible symptoms; HR, highly resistant and sometimes with hypersensitive reactions to infections.

**Table 2 plants-12-00421-t002:** Host susceptibility of *Colletotrichum echinochloae* in major weedy plants in Hubei Province.

Family	Plant Species	Plant Reaction
*Amaranthaceae*	*Celosia argentea* L.	I
	*Alternanthera philoxeroides* (*Mart.*) *Griseb.*	I
	*Achyranthes bidentata Blume*	I
*Chenopod*	*Chenopodium ficifolium Sm.*	I
*Convolvulaceae*	*Convolvulus arvensis* L.	I
*Malvaceae*	*Abutilon theophrasti Medikus*	I
*Polygonaceae*	*Persicaria hydropiper* (L.) *Spach*	I
*Portulacaceae*	*Portulaca oleracea* L.	I
*Rubiaceae*	*Scleromitrion diffusum* (*Willd.*) *R.J. Wang*	I
*Scrophulariaceae*	*Lindernia crustacea* (L.) *F. Muell.*	I
*Solanaceae*	*Solanum nigrum* L.	I
*Violaceae*	*Viola japonica Langsd. ex DC.*	I
*Asteraceae*	*Eclipta prostrata* (L.) L.	I
	*Bidens Pilosa* L.	I
	*Bidens frondose* L.	I
	*Erigeron canadensis* L.	I
	*Symphyotrichum subulatum* (*Michx.*) *G. L. Nesom*	I
*Commelinaceae*	*Commelina communis* L.	I
	*Commelina benghalensis* L.	I
*Cyperaceae*	*Cyperus rotundus* L.	HR
*Gramineae*	*Digitaria sanguinalis* (L.) *Scop.*	I
	*Eleusine indica* (L.) *Gaertn.*	I
	*Leptochloa chinensis* (L.) *Nees*	I
	*Setaria viridis* (L.) *P. Beauv.*	I
	*Leptochloa panicea* (*Retz.*) *Ohwi*	I

I, plants are immune and showed no visible symptoms; HR, highly resistant and sometimes with hypersensitive reactions to infections.

**Table 3 plants-12-00421-t003:** Effect of the concentration of the *C. echinochloae* B-48 inoculum on weed control.

Conidial Concentration	Fresh Weight Inhibition Rate (%)	Disease Severity (%)
0	0 ± 0 d	0 ± 0 e
10^4^	10.07 ± 5.79 c	8.58 ± 0.09 d
10^5^	23.96 ± 0.59 b	27.99 ± 0.11 c
10^6^	34.43 ± 1.71 b	62.50 ± 0.05 b
10^7^	78.34 ± 3.78 a	73.87 ± 0.04 ab
10^8^	84.33 ± 6.50 a	78.96 ± 0.02 a

The lower-case letter represents significant difference at *p* < 0.05.

**Table 4 plants-12-00421-t004:** Efficacy of *C. echinochloae* B-48 in controlling barnyardgrass at different growth stages.

Leaf Stage	Fresh Weight Inhibition Rate (%)	Disease Severity (%)
4	79.89 ± 2.66 a	89.67 ± 0.88 a
5	56.34 ± 3.69 b	83.66 ± 3.17 a
6	57.80 ± 2.37 ab	66.34 ± 4.25 b
7	28.91 ± 3.89 c	64.67 ± 3.48 b
8	27.69 ± 2.10 c	36.16 ± 2.60 c
9	24.71 ± 4.47 c	15.32 ± 4.25 d

The lower-case letter represents significant difference at *p* < 0.05.

**Table 5 plants-12-00421-t005:** Efficacy of *C. echinochloae* B-48 in controlling barnyardgrass at different wetness periods.

Wetness Period (h)	Fresh Weight Inhibition Rate (%)	Disease Severity (%)
0	1.27 ± 1.07 d	0 ± 0 d
6	10.51 ± 7.17 c	11.67 ± 3.51 c
8	21.93 ± 13.09 c	32.38 ± 4.73 b
10	48.34 ± 6.73 b	78.33 ± 4.04 a
12	68.54 ± 5.09 a	82.67 ± 4.16 a
24	79.50 ± 3.01 a	84.32 ± 4.72 a

The lower-case letter represents significant difference at *p* < 0.05.

## Data Availability

Not applicable.

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
