# Peer review of "Colletotrichum echinochloae*: A Potential Bioherbicide Agent for Control of Barnyardgrass (*Echinochloa crus-galli* (L.) Beauv.)"

_plants, 2023, doi:10.3390/plants12030421_

Round 1

Reviewer 1 Report

The manuscript of Gu et al. “Colletotrichum echinochloae: a potential bioherbicide agent for 2 control of barnyardgrass (Echinochloa crus-galli (L.) Beauv.)” devoted to a preliminary evaluation of a weed pathogen from the genus Colletotrichum as a potential mycoherbicide. Indeed, some Colletotrichum spp. are known as weed pathogens and some strains were registered as mycoherbicides. Barnyardgrass is a problem weed and a target for development of bioherbicides. The authors found an interesting strain and characterized it. However, they need some improvements in the manuscript to make their research publishable.  

1. Abstract should be checked for repeated words, “barnyardgrass shoots”.

2. In the section 2.1 morphological features were not described fully: more media should be used, more sequences should be analyzed (ITS only is not enough for species identification, see https://doi.org/10.3114/sim0011, https://doi.org/10.3114/sim0014). Photos presented can characterize other Colletotrichum species not only Colletotrichum echinochloae: discuss what differences with other species from barnyardgrass. Moreover, the description of isolate B-48 is not similar to  Moriwaki, J., & Tsukiboshi, T. (2009). Colletotrichum echinochloae, a new species on Japanese barnyard millet (Echinochloa utilis). Mycoscience, 50(4), 273–280. doi:10.1007/s10267-009-0485-1 

3. Conidial germination at 35C may be unusual and should be checked or discussed comparing with literature data.

4. In the section 2.5. table of figure data should be presented.

5. Inoculation experiments were performed at 48-h leaf wetness period. It is too long for true pathogens. This period may be appropriate for host range study but should be shorter in the case study of effect of conidial concentration and plant stage. Moreover, the effect of leaf wetness period on disease severity should be evaluated for characterization of the strain as a strong pathogen.

6. More literature data should be used in the discussion, authors should refer to tables and figures in the Results section.

7. ANOVA or other test should be used for statistics.

Reviewer 2 Report

1.      How dew duration, dew period temperature, plant growth stage, pathogen concentration, and adjutants affect bioherbicide efficacy? In the 5th paragraph of Introduction, the factor about dew period temperature and pathogen concentration were not mentioned.

2.      Are there precedents for mycoherbicide C. echinochloae? Why you choose this mycoherbicide? It is somewhat stiff to put C. echinochloae as the research object of the article directly in the last paragraph of Introduction.

3.      The expression about “the bars indicate 10 µm” is inaccurate, is it indicate the scaling? And there is a little detail error that missing a dot before "(c)" in the title of Figure 1.

4.      Please supplement notes about meaning of HR and I in the table.

5.      Please increase the pixels of the picture.

6.      The references of C. echinochloae B-48 inoculum concentration?

7.      What is PSB in line 292? Please add the full words.

8.      How did you assess the treatment effects of C. echinochloae on host?

9.      Where did you reference from about liquid formulation in line 345?

10.   Were all data satisfy the assumptions of normality and homogeneity of variance before ANOVA? And what factors were assessed in ANOVA?

11.   Please complement the treatment differences according to LSD which were not displayed in figure 2-3.

12.   In line 126-127, it showed that “All the plant species tested were immune”. However, Cyperus rotundus L. appeared to be a hypersensitive reaction in table 2.

13.   Did you consider the negative impact of high concentration of herbicides on the environment? And what is the appropriate application concentration of the mycoherbicide in agriculture management?

14.   Where are the data about effect of C. echinochloae B-48 on different weed growth stage? Such as fresh weight you mentioned. And what indicators does the inhibition rate refer to?

15.   In Line 198-200 of Discussion, it mentioned that the difficulty of the genus Colletotrichum products have failed to become widely available due to the expensiveness and small niche range. Why you still develop C. echinochloae as the potential herbicide? What is the significance?

Round 2

Reviewer 1 Report

Line 17: delete “very”

Line 36: edit “repeated use of herbicides can induce resistance to herbicides”

Line 42: change “great”

Line 60-62 Ref 37 38 – delete inconsistent sentence

Line 82-83: the aim concerns your isolate only.

Please, set the different figure 2 for a phylogenetic tree

Line 131 edit

Line 145-150 – discussion or introduction

Line 173-174 do not use decimals at means (here and after)

Table 3 do not use decimals and errors at zero

Check title of the Table 5

English should be checked
